# RE-MEANFLOW: EFFICIENT ONE-STEP GENERATIVE MODELING VIA MEANFLOW ON RECTIFIED TRAJECTORIES

## ABSTRACT

Flow models have demonstrated remarkable capabilities in generative modeling, yet a key bottleneck is the expensive numerical integration of ODEs required for sampling. Rectified Flow mitigates this by iteratively learning straighter trajectories through a reflow mechanism, but repeated rounds of training incur heavy computation overhead and often degrade sample quality. More recently, Meanflow has shown strong one-step generation by directly modeling the average velocity across time, but training it from scratch is costly, as it must learn from noisy signals induced by highly curved flows. To address these limitations, we propose **Re-Meanflow**, which trains a Meanflow model on trajectories straightened once using a single reflow step. The key insight is that "reflow" alone is too costly to achieve nearly straight paths for one-step sampling, while Meanflow can tolerate less-straight trajectories but is prohibitively expensive to train from scratch. By combining them, Re-Meanflow leverages their complementary strengths: a single reflow step produces sufficiently straight trajectories, enabling efficient Meanflow training without the performance degradation of repeated reflow processes. We evaluate Re-Meanflow on ImageNet $64^2$ and $256^2$, where it achieves competitive or superior one-step generation compared to state-of-the-art methods while offering substantial training efficiency. In particular, on ImageNet $64^2$, our method improves the FID of 2-rectified flow++ by $33.4\%$ while reducing training cost by $90\%$.

## 1 INTRODUCTION

Flow models (Lipman et al., 2022; Liu et al., 2022) and closely related diffusion models (Song & Ermon, 2019; Sohl-Dickstein et al., 2015) have become a central paradigm in generative modeling, enabling a wide range of applications across various data domains (Ho et al., 2022; Rombach et al., 2022; Zhang et al., 2025). Compared with earlier paradigms such as GANs (Goodfellow et al., 2020; Karras et al., 2019) and Normalizing Flows (Rezende & Mohamed, 2015; Zhai et al., 2024), these models offer stable training and superior fidelity, but at the cost of expensive sampling: high-quality generation typically requires dozens of neural function evaluations (NFEs). This computational overhead remains a central barrier to deploying flow-based generative models at scale.

The root cause of this inefficiency is the curvature of the generative trajectories induced by the prior and data distributions. In practice, the velocity fields governing these flows bend sharply, making them difficult to approximate with only a few discretization steps. Even if the instantaneous velocity is perfectly modeled, a single Euler step cannot follow the curved path and instead collapses the predictions towards the mean of plausible outcomes (Frans et al., 2024). Advanced ODE/SDE solvers (Song et al., 2020a; Karras et al., 2022; Lu et al., 2022; Zhang & Chen, 2022) reduce the discretization error, but still require 10-20 steps to achieve acceptable fidelity, leaving the generation of one step predictions out of reach.

Several strategies have been proposed to reduce the sampling cost of flow-based generative models. Rectified Flow (Liu et al., 2022; Lee et al., 2024) addresses this limitation by progressively straightening trajectories through reflow, thereby reducing the number of sampling steps. However, it requires retraining and often degrades quality. While Lee et al. (2024) claims that performing two rounds of rectification already yields sufficiently straight trajectories, our theoretical analysis and toy examples demonstrate that noticeable curvature remains even after double rectification, suggesting that further challenges may arise when building upon such trajectories. In contrast, Meanflow (Geng et al., 2025) bypasses ODE integration by directly modeling average displacements, achieving strong empirical results even with single-step sampling. However, training remains both expensive and unstable because the supervision signals are noisy when flows are highly curved. Furthermore, when extended to classifier-free guidance (CFG), Meanflow must be trained directly on the CFG-modified flow, which requires additional parameter tuning and further increases the computational burden.

In this work, we propose **Re-Meanflow**, a conceptually simple yet computationally efficient two-stage approach that combines the strengths of Rectified Flow and Meanflow while addressing their respective weaknesses. In the first stage, we apply a single reflow step on a pretrained flow model to obtain trajectories that are significantly straighter. In the second stage, we train a Meanflow model on these trajectories. This combination yields clear benefits: rectification reduces the curvature that complicates numerical integration, while Meanflow removes the need for integration altogether by directly learning average displacements. As a result, Re-Meanflow learns from velocity fields that are already close to straight, providing a cleaner and more stable training signal and substantially simplifying the learning problem compared to training Meanflow directly on unrectified flows.

We show that Re-Meanflow substantially improves both training efficiency and generation quality. On ImageNet, we evaluate $64^2$ resolution in pixel space initialized from EDM2 (Karras et al., 2024b) and $256^2$ resolution in latent space initialized from SiT (Ma et al., 2024). In both settings, Re-Meanflow achieves competitive one-step generation quality while offering significant gains in efficiency. On ImageNet $64^2$, compared to 2-rectified flow++ (Lee et al., 2024), our method reduces FID by $33.4\%$ while using only $10\%$ of the total computation. We further surpass recent state-of-the-art distillation approaches such as sCD (Lu & Song, 2024) and AYS (Sabour et al., 2025), achieving both higher quality and lower cost. On ImageNet $256^2$, Re-Meanflow achieves comparable performance to Meanflow (Geng et al., 2025) trained from scratch, but with significantly higher efficiency. In contrast, finetuning Meanflow from the same initialized model yields only about 10 FID under the same compute budget.

## 2 RELATED WORKS

**Flow and diffusion-based generative models.** Diffusion-based generative models (Sohl-Dickstein et al., 2015; Song & Ermon, 2019; Ho et al., 2020; Song et al., 2020b) learn to reverse a gradual noising process, where the reverse-time dynamics can be formulated either as a stochastic SDE or a deterministic probability-flow ODE (Song et al., 2020b; Karras et al., 2022). Flow Matching methods (Lipman et al., 2022; Albergo & Vanden-Eijnden, 2022; Liu et al., 2022) generalize this perspective by directly regressing velocity fields that transport mass between source and target distributions, establishing close connections to continuous-time normalizing flows (Rezende & Mohamed, 2015). Although these models achieve high fidelity, iterative ODE/SDE integration remains computationally expensive. Recent work has accelerated sampling with improved numerical solvers (Song et al., 2020a; Karras et al., 2022; Lu et al., 2022; Zhang & Chen, 2022), yet the high curvature of generative paths still hinders few-step sampling. Rectified Flow methods (Liu et al., 2022; Tong et al., 2023; Lee et al., 2024) alleviate this by explicitly learning straighter trajectories that enable one-step or few-step sampling, but such straight-path construction itself incurs significant computational cost due to heavy training or repeated reflow procedures. In this work, we will address this tradeoff by retaining the efficiency benefits of Rectified Flow while avoiding its expensive path-construction process.

**Accelerated Sampling.** Reducing sampling steps has been central to making diffusion models practical. Distillation approaches compress long diffusion chains into few-step models (Salimans & Ho, 2022; Geng et al., 2025; Sauer et al., 2024), including score-based distillation methods (Luo et al., 2023; Yin et al., 2024; Zhou et al., 2024). A distinct line eliminates distillation entirely: Consistency Models train networks to produce invariant outputs across different timesteps, enabling direct few-step generation (Song et al., 2023; Song & Dhariwal, 2023; Geng et al., 2024; Lu & Song, 2024; Yang et al., 2024). More recent work revisits the formulation of time evolution itself-for example, Flow Maps (Boffi et al., 2024), Shortcut Models (Frans et al., 2024), and Inductive Moment Matching (Zhou et al., 2025) - all of which target few-step efficiency but often struggle with stability or high training cost. Also, training these models is challenging, as they must learn flow maps along curved trajectories rather than follow smooth diffusion paths. Recent work has addressed this by simplifying the consistency objective (Lu & Song, 2024) and introducing improved loss functions and normalization strategies for latent consistency models (Dao et al., 2025). Our Re-Meanflow builds on these insights, combining trajectory simplification with robust training process to achieve both training efficiency and one step generation quality.

## 3 BACKGROUND

### 3.1 RECTIFIED FLOW AND REFLOW (LIU ET AL., 2022).

Given a prior distribution $p_{\mathbf{z}}$ (taken to be $\mathcal{N}(\mathbf{0}, \mathbf{I})$ in this work) and a data distribution $p_{\mathbf{x}}$, Rectified Flow formulates a generative model that transports $p_{\mathbf{z}}$ to $p_{\mathbf{x}}$ through an ordinary differential equation (ODE) over time $t \in [0, 1]$. For a data-noise coupling $(\mathbf{x}, \mathbf{z}) \sim p_{\mathbf{xz}}$, where $p_{\mathbf{xz}}$ denotes the joint distribution of $\mathbf{x}$ and $\mathbf{z}$, the flow path is defined as a linear interpolation $\mathbf{z}_t = (1 - t)\mathbf{x} + t\mathbf{z}$ with the **conditional velocity** $v_t = d\mathbf{z}_t/dt = \mathbf{z} - \mathbf{x}$. The flow model $v_\theta$ is trained to match the **marginal velocity field** $v_t^\star(\mathbf{z}_t, t) = \frac{1}{t}(\mathbf{z}_t - \mathbb{E}[\mathbf{x} \mid \mathbf{x}_t = \mathbf{z}_t])$ by regressing the conditional velocity of observed couplings:

$$\mathcal{L}_{MF}(\theta) = \mathbb{E}_{\mathbf{x}, \mathbf{z} \sim p_{\mathbf{xz}}, \, t \sim p_t}[||(\mathbf{z} - \mathbf{x}) - v_\theta(\mathbf{z}_t, t)||_2^2]. \tag{1}$$

Once $v_\theta$ is learned, a new sample $\mathbf{x}$ can be generated by solving the ODE for $\mathbf{z} \sim p_{\mathbf{z}}$:

$$\mathbf{x}_\theta(\mathbf{z}) = \mathbf{z}_0 = \mathbf{z} - \int_0^1 v_\theta(\mathbf{z}_\tau, \tau)d\tau. \tag{2}$$

Rectified Flow begins training with independently sampled $\mathbf{x}$ and $\mathbf{z}$, i.e., $p_{\mathbf{xz}}^0(\mathbf{x}, \mathbf{z}) = p_{\mathbf{x}}(\mathbf{x})p_{\mathbf{z}}(\mathbf{z})$, which we refer to as **independent coupling**. Training on these independent couplings produces highly curved ODE trajectories, since the flow paths of different data-noise pairs become interlaced and the model only learns the marginal velocity. Such curved trajectories, in turn, require a large number of function evaluations (NFEs) to accurately solve Eq. 2 with numerical ODE solvers. To mitigate this, Rectified Flow adopts a **reflow** process that proceeds iteratively. Given a coupling $p_{\mathbf{xz}}^k$, we first sample training pairs $(\mathbf{x}, \mathbf{z})$ from it and learn a new vector field $v_{\theta^{k+1}}$. This vector field is then used to solve Eq. 2 and construct the updated coupling $p_{\mathbf{xz}}^{k+1}$. For clarity, we refer to the trained vector field $v_{\theta^{k+1}}$ as the $(k+1)$-**rectified flow**. For example, starting from the independent coupling $p_{\mathbf{xz}}^0$, the first trained vector field - though typically still highly curved - is what we call the 1-rectified flow. Liu et al. (2022); Lee et al. (2024) show that 2-rectified flow is significantly straighter, allowing the velocity model trained on this flow path to sample with few steps.

### 3.2 MEANFLOW (GENG ET AL., 2025).

Compared to Rectified Flow (Liu et al., 2022), which models the *instantaneous velocity* and requires solving the numerical ODE in Eq. 2 during sampling, Meanflow enables one-step sampling by modeling a new field

representing the *average velocity* $u$:

$$u(\mathbf{z}_t, r, t) \triangleq \frac{1}{t-r} \int_r^t v(\mathbf{z}_\tau, \tau) d\tau. \tag{3}$$

The objective is to approximate this average velocity with a neural network $u_\theta(\mathbf{x}_t, r, t)$, so that the entire flow path can be reconstructed from a single evaluation $u_\theta(\mathbf{z}, 0, 1)$, directly replacing the integral term in Eq. 2. The key idea of Meanflow is to transform the intractable integral into a learnable target by taking the time derivative of both sides of Eq. 3:

$$\underbrace{u(\mathbf{z}_t, r, t)}_{\text{average vel.}} = \underbrace{v(\mathbf{z}_t, t)}_{\text{instant. vel}} - (t-r) \underbrace{\frac{d}{dt} u(\mathbf{z}_t, r, t)}_{\text{time derivative}}. \tag{4}$$

This identity allows Meanflow to bypass direct integral computation and exploit the right-hand side as the implicit training signal for $u_\theta$. Specifically, the training objective is:

$$\mathcal{L}_{MF}(\theta) = \mathbb{E}_{\mathbf{x},\mathbf{z},r,t} ||u_\theta(\mathbf{z}_t, r, t) - \text{sg}(u_{\text{tgt}})||_2^2 \tag{5}$$

$$u_{\text{tgt}} = v(\mathbf{z}_t, t) - (t-r)\frac{d}{dt} u_\theta(\mathbf{z}_t, r, t). \tag{6}$$

Here, $\text{sg}(\cdot)$ denotes the stop-gradient operator, and the derivative $\frac{d}{dt} u(\mathbf{z}_t, r, t)$ can be decomposed as $v(\mathbf{z}_t, t) \partial_{\mathbf{z}} u + \partial_t u$, which can be computed using the Jacobian-vector product (JVP) interface in frameworks such as PyTorch or JAX. Meanflow also naturally extends to classifier-free guidance (CFG) by replacing the velocity field $v$ in Eq. 6 with the CFG-enhanced velocity $v^{\text{cfg}}$:

$$v^{\text{cfg}}(\mathbf{z}_t, t \mid c) \triangleq \omega \, v(\mathbf{z}_t, t \mid c) + (1 - \omega) \, v(\mathbf{z}_t, t). \tag{7}$$

## 4 METHOD

### 4.1 MOTIVATION

Rectified Flow and Meanflow both aim to reduce the high sampling cost caused by curved ODE trajectories, but they do so in different ways and face complementary challenges. Importantly, combining them yields a *synergistic* effect rather than a simple sum of benefits.

**Rectified Flow.** By explicitly straightening the trajectories, Rectified Flow enables few-step sampling. Lee et al. (2024) argue that intersections between two flow paths $(\mathbf{x}', \mathbf{z}')$ and $(\mathbf{x}'', \mathbf{z}'')$ are exceedingly rare. Their reasoning is that if two paths intersect at time $t$, then $\mathbf{z}'' = \mathbf{z}' + \frac{1-t}{t}(\mathbf{x}' - \mathbf{x}'')$, which would place $\mathbf{z}''$ far outside the true noise distribution unless $\mathbf{x}'$ and $\mathbf{x}''$ are extremely close. Hence, they conclude that intersections almost never occur, and the resulting marginal velocity field should already be sufficiently straight. They further acknowledge that intersections are likely to occur when $t$ is close to 1, since in this regime $\frac{1-t}{t}$ becomes vanishingly small and $\mathbf{z}''$ remains close to $\mathbf{z}'$ regardless of $\|\mathbf{x}' - \mathbf{x}''\|$. To justify straightness in this edge case, Lee et al. (2024) assumes that the 1-rectified flow is $L$-Lipschitz, which would imply that $x'$ and $x''$ are themselves close and that linear interpolation between them should remain near the data manifold. However, this assumption is overly strong. In practice, $x'$ and $x''$ can still be separated in high-dimensional space, and real data often lie on thin, non-convex manifolds. As a result, their average or any straight-line interpolation between them does not necessarily remain on the data manifold. Consistent with our analysis, a toy example (Fig. 2) confirms that flows indeed bend near $t = 1$, causing single-step Euler sampling to drift into invalid regions and generate outlier samples. This highlights the need for alternative strategies to address residual curvature in flow paths, rather than assuming that double rectification is sufficient.

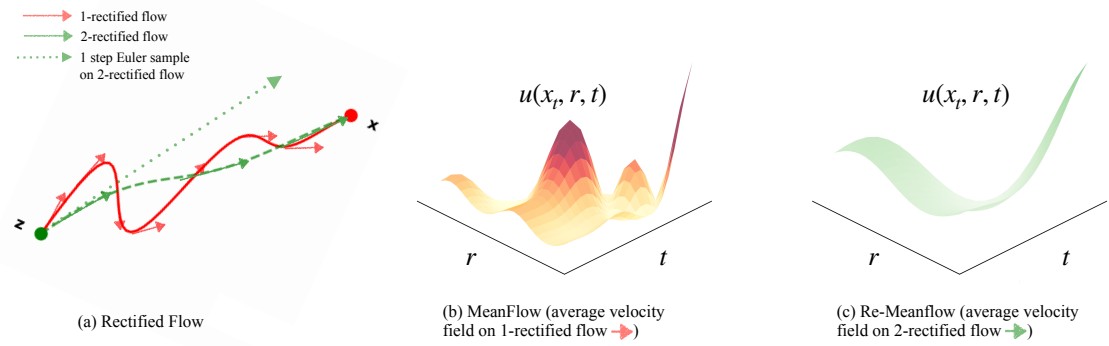

Figure 1: **Illustration of the motivation in Sec. 4**. (a) 1-rectified flow (→) follows highly curved trajectories, requiring many ODE steps. Applying two rounds of rectification (→) straightens the paths and reduces the NFEs, but one-step sampling (··▸) still fails unless trajectories are nearly perfectly straight. (b) The average velocity field from the 1-rectified flow: Meanflow models $u(\mathbf{z}_t, r, t)$ over all intervals $(r, t)$, which in principle bypasses the need for perfectly straight paths. However, when the underlying velocity field is curvy, the induced averages are complex and lead to slow convergence. (c) The average velocity field from the 2-rectified flow: training Meanflow on such reflowed trajectories yields a much cleaner and smoother field, making estimation easier and enabling faster convergence with improved one-step generation.

**Meanflow.** Training Meanflow directly on curved flows suffers from noisy and unstable supervision signals, which is a well-known cause of slow convergence in machine learning. Concretely, in the training objective of Eq. 6, the marginal velocity field is not directly available and must be approximated from conditional velocities. Because the couplings are sampled independently, these conditional velocities exhibit high variance. Moreover, the derivative term $(t - r) \frac{d}{dt} u(\mathbf{z}_t, r, t)$ relies on Jacobian-vector products of the model output without explicit supervision, introducing additional noise that compounds the variance. Finally, when trajectories are highly curved, the corresponding average velocity field becomes irregular and difficult to approximate (Fig. 1b).

**Synergy.** Although Rectified Flow straightens trajectories, a single round of rectification is not sufficient: residual curvature near $t = 1$ still undermines reliable single-step sampling. By coupling Rectified Flow with Meanflow, we address this limitation, as Meanflow learns average velocities on already-straightened paths. This combination further mitigates the main challenges of training Meanflow directly: 1) In straighter flows, path intersections are greatly reduced, so conditional velocities provide a closer approximation to the marginal velocity field; 2) Since instantaneous velocities are closer to their averages, the discrepancy between $u(\mathbf{z}_t, r, t)$ and $v(\mathbf{z}_t, t)$ in Eq. 5 is reduced, leading to more stable Jacobian-vector product estimates; 3) Rectified trajectories yield average velocity fields that are easier to approximate (Fig. 1c). Together, these effects demonstrate the complementary strengths of Rectified Flow and Meanflow.

## 4.2 RE-MEANFLOW

Given a pretrained flow model $v_\theta$ trained on the original independent couplings $p_{\mathbf{xz}}(\mathbf{x}, \mathbf{z}) = p_{\mathbf{x}}(\mathbf{x})p_{\mathbf{z}}(\mathbf{z})$, we construct a new coupling distribution $p_{\mathbf{xz}}^1$ using its learned velocity field. Couplings can be obtained in two ways: (i) by sampling a noise vector $\mathbf{z}$ and solving Eq. 2 forward to obtain the corresponding $\mathbf{x}$, or (ii) by sampling a data point $\mathbf{x}$ and integrating the flow backward to its corresponding $\mathbf{z}$. The flow induced by $p_{\mathbf{xz}}^1$ yields trajectories that are noticeably straighter, with reduced curvature.

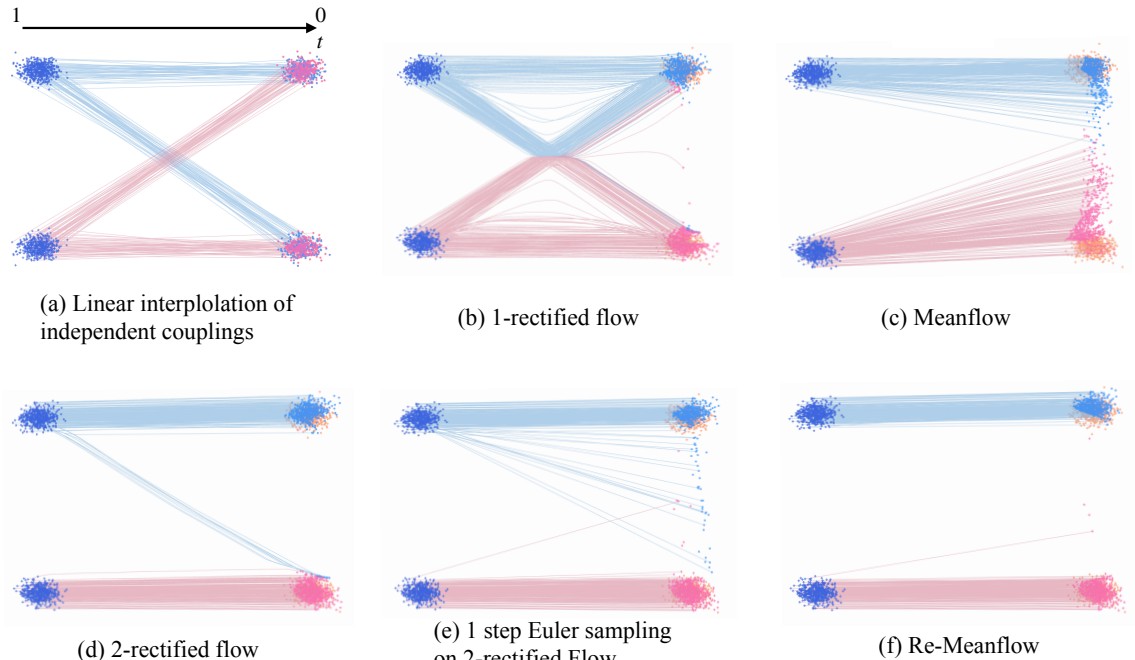

Figure 2: **Toy experiment on a 2D Gaussian mixture task with fixed training budget.** (a) Linear interpolation of independent couplings. (b) 1-rectified flow, the learned flow after training on linear interpolations of independent couplings, exhibiting high curvature, which requires high NFE to simulate. (c) 2-rectified flow obtained after a reflow step on (b), which reduces but does not fully remove curvature. (d) One-step Euler sampling on the 2-rectified flow produces many outliers because the trajectories are not perfectly straight. (e) Meanflow trained directly on independent couplings fails to capture the flow within the limited training budget. (f) Meanflow trained on a 1-rectified flow achieves accurate one-step generation with minimal outliers.

We finetune the pretrained flow model into a Meanflow model $u_\theta$ under the Meanflow objective (Eq. 5). Because Meanflow does not require strictly straight trajectories, training on $p_{\mathbf{xz}}^1$ converges faster and more stably than training a conventional flow model. However, some $(\mathbf{x}, \mathbf{z})$ pairs are obtained by solving Eq. 2 starting from real images $\mathbf{x}$, in which case the resulting $\mathbf{z}$ may deviate from the true Gaussian prior. To mitigate this misalignment and to avoid $\mathbf{z}$ collapsing onto discrete points, we inject small Gaussian perturbations into $\mathbf{z}$ during finetuning. Specifically, to better preserve the Gaussian prior, each $\mathbf{z}$ is updated as a convex combination of the original noise and fresh Gaussian noise:

$$\mathbf{z}' = \sqrt{1-\rho}\,\mathbf{z} + \sqrt{\rho}\,\epsilon, \quad \epsilon \sim \mathcal{N}(0, I), \tag{8}$$

where $\rho \in [0, 1]$ denotes the noise ratio.

To incorporate classifier-free guidance (CFG) (Ho & Salimans, 2022), we avoid directly training on the CFG velocity field $v^{\mathrm{cfg}}$ (Eq. 7), which tends to be unstable and prone to collapse. Instead, we adopt a two-stage strategy: first train $u_\theta$ on the unconditional flow, and then perform a brief finetuning stage on the CFG-modified flow. Empirically, allocating 80% of training iterations to the unconditional flow and the remaining 20% to CFG yields stable convergence and allows flexible tuning. Once trained on the unconditional flow, CFG weights can be adjusted efficiently at inference time.

Table 1: **Class-conditional generation on ImageNet** $64^2$ **(left) and** $256^2$ **(right).** All results are reported with classifier-free guidance (CFG) for methods that support it. "$\times 2$" denotes that CFG doubles the NFE per step. For distillation-based methods, we also report training compute in estimated exaFLOPs (Eflops) as a standardized measure of efficiency.

| METHOD | NFE ($\downarrow$) | FID ($\downarrow$) | Train + (Sample) Eflops ($\downarrow$) |
|---|---|---|---|
| **Diffusion models** | | | |
| ADM (Dhariwal & Nichol, 2021) | 250×2 | 2.07 | |
| EDM (Karras et al., 2022) | 63×2 | 2.30 | |
| EDM2-S (Karras et al., 2024b) | 63×2 | 1.58 | |
| **Score distillation** | | | |
| DMD (Yin et al., 2024) | 1 | 2.62 | |
| EMD (Xie et al., 2024) | 1 | 2.20 | |
| **Diffusion distillation** | | | |
| CD (Song et al., 2023) | 1 | 6.20 | 811 |
| 2-rectified flow++ † (Lee et al., 2024) | 1 | 4.31 | 473 + (35) |
| sCD (Lu & Song, 2024) | 1 | 2.97 | 501 |
| AYF (Sabour et al., 2025) | 1 | 2.98 | 63 |
| **Re-Meanflow (ours)*** | **1** | **2.87** | **4 + (49)** |

| METHOD | NFE ($\downarrow$) | FID ($\downarrow$) | EFlops ($\downarrow$) |
|---|---|---|---|
| **Diffusion models** | | | |
| ADM (Dhariwal & Nichol, 2021) | 250×2 | 10.94 | |
| DiT-XL (Peebles & Xie, 2023) | 250×2 | 2.27 | |
| SiT-XL (Ma et al., 2024) | 250×2 | 2.06 | |
| **Few Step Models** | | | |
| iCT (Song & Dhariwal, 2023) | 1 | 34.6 | |
| Shortcut Model (Frans et al., 2024) | 1 | 10.6 | |
| iMM (Zhou et al., 2025) | 1×2 | 7.77 | |
| Meanflow (Geng et al., 2025) | 1 | 3.43 | |
| **Re-Meanflow (ours)** | **1** | **3.48** | **3+(194)** |

Finally, to illustrate the benefits of Re-Meanflow, we conduct a controlled 2D experiment with source and target distributions both given by mixtures of two Gaussians. The velocity and Meanflow models are parameterized by small MLPs, and the training budget is fixed at 20k iterations (batch size 1024, Adam optimizer, learning rate $10^{-3}$). We compare the one-step generation quality under three settings: (1) **2-rectified flow**, where two reflow steps are applied to independent couplings, each trained for 10k iterations; (2) **Meanflow**, trained for 20k iterations directly on independent couplings; (3) **Re-Meanflow (ours)**, where a velocity model is trained for 10k iterations to obtain a 1-rectified flow, followed by another 10k iterations of Meanflow on the resulting couplings. As shown in Fig. 2, 2-rectified flow still produces outliers under one-step Euler sampling on imperfectly straight trajectories (panel d), while Meanflow alone fails to converge within budget on curved flows (panel e). In contrast, Re-Meanflow eliminates most outliers and achieves accurate one-step generation (panel f). This example illustrates the synergy between the two components: rectification reduces curvature enough to stabilize Meanflow, while Meanflow avoids the heavy cost of fully straightening trajectories.

## 5 EXPERIMENTS

We conduct experiments on ImageNet (Deng et al., 2009) at $64^2$ resolution in pixel space and $256^2$ resolution in latent space. In both settings, we initialize Re-Meanflow from a pretrained flow or diffusion model and generate 5M couplings for the reflow process. For ImageNet-$64^2$, we initialize Re-Meanflow from the pretrained EDM2-S (Karras et al., 2024b) and reparameterize it from the original VE diffusion formulation into the flow-matching setting following Lee et al. (2024); Sabour et al. (2025). For ImageNet-$256^2$, we initialize Re-Meanflow from the pretrained SiT-XL (Ma et al., 2024). In both cases, couplings are generated using the default sampling method described in the corresponding papers. All efficient training runs are performed on 8 A100 GPUs with a global batch size of 128 for 70k iterations (50k in the first stage and 20k in the second), using an EMA decay rate of 0.9999.

### 5.1 ONE STEP GENERATION QUALITY

For both experiments, we evaluate image quality using the Fréchet Inception Distance (FID) (Heusel et al., 2017) and measure training efficiency in terms of the total floating-point operations (FLOPs). Re-Meanflow outperforms or is competitive with state-of-the-art baselines in one-step generation while offering substantial efficiency gains, demonstrating that our method achieves both strong sample quality and practical scalability.

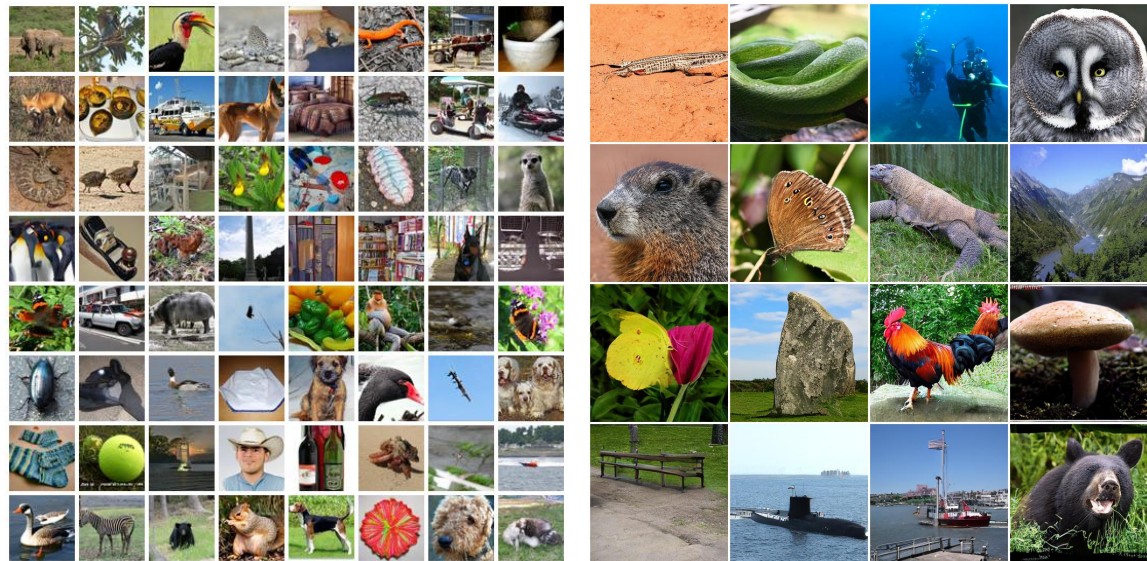

Figure 3: **Qualitative for one step sampling on on ImageNet $64^2$ (Left) and $256^2$ (Right)**.

On ImageNet $64^2$, Re-Meanflow achieves an FID of 2.87 in one step, outperforming 2-rectified flow++ (Lee et al., 2024) by 1.3 FID (a 33.4% reduction). Compared to recent state-of-the-art diffusion- or flow-based methods such as AYF (Sabour et al., 2025) and sCD (Lu & Song, 2024), all initialized from EDM2 (Karras et al., 2024b), we further improve FID by $\sim$0.1. We also close the gap against score distillation approaches, showing that one-step generation is attainable without additional distillation stages. This demonstrates that training Meanflow on a single reflowed trajectory provides a clear advantage over repeated rectification. On ImageNet $256^2$, Re-Meanflow achieves an FID of 3.48, matching the performance of Meanflow (Geng et al., 2023). Qualitative results are shown in Fig. 3.

## 5.2 TRAINING EFFICIENCY

As shown by Lee et al. (2024), distillation methods involving reflow can still be more efficient than alternative distillation approaches, despite requiring additional coupling generation. We follow a similar analysis by approximating the total training cost in FLOPs (Table 1), and we also examine the convergence behavior by tracking FID against training iterations (Fig. 4).

On ImageNet $64^2$, Re-Meanflow requires only 10% of the training cost of 2-rectified flow++ Lee et al. (2024), and 82% of the cost of the second-most efficient method, AYF Sabour et al. (2025). Moreover, after generating the couplings (a one-time cost), training itself accounts for just 6% of the total, leaving more compute available for hyperparameter exploration (e.g., noise schedule, CFG strength).

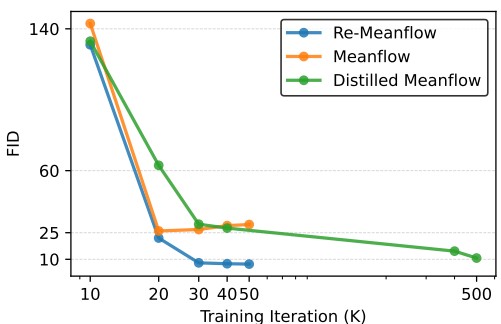

Figure 4: Convergence behavior comparison between Meanflow and Re-Meanflow.

On ImageNet $256^2$, vanilla Meanflow fails to converge, plateauing at FID $\sim$25 with a batch size of 64 (the maximum supported by our 8 GPUs). We therefore introduce *Distilled Meanflow*, which replaces the

conditional velocity in Eq. 6 with a pretrained marginal velocity, yielding more stable training and aligning with our intuition in Sec. 4. Nevertheless, Re-Meanflow converges both faster and to a higher final quality than either Meanflow or Distilled Meanflow, even under the same training budget (Fig. 4).

## 5.3 ABLATIONS

**Effect of coupling configurations.** We study the effect of different coupling configurations (Table 2) by comparing four setups: (i) *real pairs*, formed by real images and their corresponding noise vectors recovered by solving Eq. 2 backward from $\mathbf{x}$; (ii) *sync pairs*, where Gaussian noise is paired with synthetic data generated by solving Eq. 2 forward from $\mathbf{z}$; (iii) a mixture of 2.6M sync pairs and 2.4M real pairs; and (iv) 5M sync pairs.

Table 2: **Ablation on coupling configurations** (ImageNet $256^2$).

| Couplings | FID ($\downarrow$) |
|---|---|
| 2.4M real pairs | 4.35 |
| 2.4M sync pairs | 4.89 |
| 2.6M sync + 2.4M real pairs | **3.48** |
| 5M sync pairs | 3.55 |

With the same number of pairs, we observe that including real pairs yields a better FID, suggesting that synthetic couplings may deviate from the true data distribution and thus reduce quality. We also observe that increasing the total number of pairs appears to improve performance, which could indicate that the benefit of broader coverage outweighs the potential degradation from lower-quality synthetic couplings.

**Effect of noise injection.** Having introduced noise injection in Eq. 8, we now evaluate its impact in different settings (Table 3). As defined in Eq. 8, each noise vector is perturbed by mixing it with fresh Gaussian noise in a variance-preserving manner. A small noise ratio of $0.1$ yields the best FID, indicating that mild perturbations enlarge the support while maintaining correspondence between noisy and clean samples. In contrast, larger ratios (e.g., $0.7$) or uniformly sampled noise break this correspondence and thus degrade image quality.

Table 3: **Ablation on noise injection** (ImageNet $256^2$).

| Noise ratio $\rho$ | FID ($\downarrow$) |
|---|---|
| $\rho = 0$ | 3.74 |
| $\rho = 0.1$ | **3.48** |
| $\rho = 0.7$ | 3.96 |
| $\rho \sim Uniform[0, 1]$ | 4.34 |

## 6 CONCLUSION

We presented **Re-Meanflow**, a simple and efficient framework that combines the complementary strengths of Rectified Flow and Meanflow. By training Meanflow on trajectories straightened once through reflow, our method both avoids the heavy retraining cost and quality degradation of iterative rectification and enables more stable training by providing cleaner signals than directly learning from curved flows.

Empirically, Re-Meanflow achieves substantial gains in both efficiency and generation quality. On ImageNet $64^2$, it delivers a 33.4% FID reduction over 2-rectified flow++ while requiring only 10% computation, and it surpasses recent state-of-the-art distillation methods such as sCD (Lu & Song, 2024) and AYS (Sabour et al., 2025) in both quality and cost. On ImageNet $256^2$, it matches the performance of Meanflow trained from scratch with great efficiency.

Nonetheless, Re-Meanflow inherits certain limitations. Because reflow generates synthetic couplings, the final performance is bounded by their quality, which may restrict further improvements. Moreover, while effective, our method remains behind score distillation approaches such as DMD (Yin et al., 2024) and EMD (Xie et al., 2024). Looking ahead, we believe that leveraging straighter trajectories is a broadly applicable principle, and extending this idea to other few-step paradigms, such as consistency models (Song et al., 2023) or shortcut diffusion (Frans et al., 2024).

## REPRODUCIBILITY STATEMENT

The models and datasets adopted in this study are described in Sec. 5. The complete training procedure together with all hyperparameter settings are reported in Appendix A. We will release all the code for data preprocessing and experimental evaluation upon publication of the paper. The code will be made available under a license that permits free use for research purposes.

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

## A    TRAINING PROCEDURES AND HYPERPARAMETER SETTINGS

**ImageNet-**$64^2$**.**  We initialize Re-Meanflow from the pretrained EDM2-S model (Karras et al., 2024b). Following Lee et al. (2024), we convert the original VE diffusion parameterization into the flow-matching setting, where the noise level is defined as $\sigma_t = \frac{t}{1-t}$. To adapt the pretrained time embeddings, we follow AYS (Sabour et al., 2025) to replace the original embedding $\text{embed}_m(\log \sigma_t)$ with a new embedding $\text{embed}_{\text{new}}(t)$, which is trained for 10,000 iterations (a few minutes of fine-tuning) to reproduce the outputs of the original module at the corresponding noise levels. For the Meanflow extension, we introduce an additional $r$ embedding, which is zero-initialized so that the initial behavior of the model matches the pretrained baseline.

The couplings are constructed by simulating the ODE with the EDM2 Heun sampler (32 steps), and auto-guidance (Karras et al., 2024a) is applied to generate high-quality synthetic images. During training, we set $p = 0.5$ in the Meanflow loss, which has a similar effect to Pseudo-Huber regularization. In the second stage of training for the CFG flows, the guidance strength is sampled uniformly from $[1, 4]$. The ImageNet dataset contains 1.2M images; to construct 2.4M "real" $(\mathbf{x}, \mathbf{z})$ pairs, we reverse noise from clean images using the default EDM2 sampling procedure without applying CFG. The time schedule follows the original EDM2 setting but is consistently converted to the $\sigma_t$ parameterization.

**ImageNet-**$256^2$**.**  We initialize Re-Meanflow from the pretrained SiT-XL model (Ma et al., 2024), which is a flow-based model, so only the additional $r$ embedding needs to be introduced. The couplings are generated using the default Heun method with 250 steps, and the synthetic couplings are produced with CFG strength 1.5. In the second stage of training for the CFG flows, the guidance strength is sampled uniformly from $[1, 2.5]$. To apply CFG, we adopt the improved trick proposed in Karras et al. (2024a), including the corresponding re-weighting equation, and set $w = 1.0$ to determine the effective $w_\pi$ scaling.

All training is conducted on 8 NVIDIA A100 GPUs, and convergence typically requires 12-15 hours.

## B    COMPUTATION ESTIMATION OF EACH METHOD

In Tables 1, we report the efficiency in terms of estimated exaFLOPs (Eflops). To ensure comparability, we estimate total training and sampling compute for each method based on their reported FLOPs per forward pass. Specifically, we use the following assumptions:

- The FLOPs of a forward pass are reported by prior works (e.g., EDMKarras et al. (2022): 100 GFLOPs, EDM2-SKarras et al. (2024b): 102 GFLOPs, SiT-XL(Ma et al., 2024): 118.64 GFLOPs).

- The FLOPs of a backward pass are measured empirically and are approximately $2\times$ the cost of a forward pass.

- For training, the total compute is computed as:

  Total Train FLOPs = (#iters) $\times$ (batch size) $\times$ (forward + backward) $\times$ (GFLOPs per fwd).

- For sampling, the total compute is computed as:

  Total Sample FLOPs = (#samples) $\times$ (#steps) $\times$ (forward passes per step) $\times$ (GFLOPs per fwd).

**Example: Our Method on ImageNet-$64^2$.** For sampling, we require two phases:

$$\underbrace{2.4 \times 10^6}_{\text{\#samples}} \times \underbrace{63}_{\text{steps}} \times \underbrace{1}_{\text{fwd/step}} \times \underbrace{102}_{\text{GFLOPs/fwd}}$$

$$+ \underbrace{2.6 \times 10^6}_{\text{\#samples}} \times \underbrace{63}_{\text{steps}} \times \underbrace{2}_{\text{fwd/step (auto-guidance)}} \times \underbrace{102}_{\text{GFLOPs/fwd}}$$

$$\approx 4.88 \times 10^{10} \text{ GFLOPs} \approx 49 \text{ Eflops.}$$

For training, we have two stages:

$$\underbrace{50{,}000}_{\text{iters}} \times \underbrace{128}_{\text{batch}} \times \underbrace{(1+2)}_{\text{fwd+back}} \times \underbrace{102}_{\text{GFLOPs/fwd}}$$

$$+ \underbrace{20{,}000}_{\text{iters}} \times \underbrace{128}_{\text{batch}} \times \underbrace{(3+2)}_{\text{fwd+back}} \times \underbrace{102}_{\text{GFLOPs/fwd}}$$

$$\approx 3.26 \times 10^8 \text{ GFLOPs} \approx 3.2 \text{ Eflops.}$$

**Example: AYS.** AYS does not require sampling, so only training compute is considered:

$$\underbrace{50{,}000}_{\text{iters}} \times \underbrace{2048}_{\text{batch}} \times \underbrace{(3+2)}_{\text{fwd+back}} \times \underbrace{102}_{\text{GFLOPs/fwd}}$$

$$\approx 6.26 \times 10^{10} \text{ GFLOPs} \approx 62.6 \text{ Eflops.}$$

**Rounding.** For consistency, all reported Eflops are rounded to the nearest integer in the main tables.

## C  USE OF LANGUAGE MODELS

We note that large language models (LLMs) were employed solely to polish the writing of this paper. They were not used in any part of the research process, including the development of methods, design of experiments, or analysis of results. Their role was limited to improving the readability and clarity of exposition, without contributing any substantive content.

