# OpenReview forum: "Re-Meanflow: Efficient One-Step Generative Modeling via Meanflow on Rectified Trajectories"
_ICLR.cc/2026/Conference — ICLR 2026 Conference Withdrawn Submission_

### Official Review · Reviewer_fy6m · 2025-10-24

**Soundness:** 2
**Presentation:** 3
**Contribution:** 2
**Rating:** 4
**Confidence:** 4

**Summary:**

This paper proposes Re-Meanflow, a two-stage method for efficient one-step generative modeling. The authors aim to combine the strengths of Rectified Flow (which straightens trajectories) and Meanflow (which enables one-step sampling by modeling average velocity) while mitigating their respective weaknesses.

The authors argue that (1) iterative Rectified Flow (e.g., 2-rectified flow) is computationally expensive to train and still leaves residual curvature that harms 1-step Euler sampling, and (2) standard Meanflow is unstable and slow to train from scratch on the highly curved, original independent couplings.

Re-Meanflow's procedure is to:
1.  Start with a pretrained flow model (a 1-rectified flow).
2.  Use this model to generate a set of "reflowed" data-noise couplings ($p_{xz}^1$).
3.  Finetune the *same* pretrained model on these new, "straighter" couplings ($p_{xz}^1$) using the Meanflow objective.

The central hypothesis is that the aligned couplings from the 1-rectified flow provide a much cleaner and more stable training signal for the Meanflow objective, leading to fast convergence. Experiments on ImageNet $64^2$ and $256^2$ show that Re-Meanflow achieves state-of-the-art or competitive 1-step FID scores while claiming a massive 90% reduction in training compute compared to the 2-rectified flow++ baseline.

**Strengths:**

1.  **Great Empirical Quality:** The paper achieves very strong 1-step generation FID scores, notably 2.87 on ImageNet $64^2$ and 3.48 on ImageNet $256^2$. These results are competitive with or superior to several state-of-the-art baselines, including 2-rectified flow++ and AYS.
2.  **Clear Problem Motivation:** The paper does a good job (e.g., in Figure 2) of illustrating the *problems* with existing methods: 1-step Euler sampling on 2-rectified flow can still produce outliers (Fig 2d), and training Meanflow from scratch on independent couplings fails to converge well (Fig 2e). This clearly motivates the need for a hybrid approach.
3.  **Simplicity:** The final algorithm is relatively simple: take a pretrained model, generate (x, z) pairs from it, and finetune with the Meanflow loss. This simplicity is appealing.

**Weaknesses:**

1.  **Unsupported Efficiency Claim:** This is the most critical weakness. The paper's central claim of a 90% training cost reduction is based on a comparison of 4 Eflops (Re-Meanflow) vs. 473 Eflops (2-rectified flow++). This 100x discrepancy is not justified and appears to be an apples-to-oranges comparison, especially since the paper's toy experiment suggests the compute costs should be identical. This invalidates the paper's main claim of superior efficiency.
2.  **Contradictory Motivation:** The paper's motivation and method are in direct conflict. The motivation section (4.1) and Figure 1c argue for the benefits of training on the smooth *trajectories of a 2-rectified flow*. However, the method (4.2) explicitly *avoids* creating a 2-rectified flow and instead trains on the *couplings from a 1-rectified flow*. This is a significant logical flaw that confuses the paper's core premise.
3.  **Marginal Novelty:** Once the efficiency claims and confusing motivation are discounted, the core idea is simply to finetune a pretrained flow model using the Meanflow objective on self-generated data. This is an incremental engineering trick, not a fundamental new method.

**Questions:**

1.  The primary question is about the 473 Eflops vs. 4 Eflops training cost in Table 1. Both methods (Re-Meanflow and 2-rectified flow++) are described as being initialized from EDM2-S. Why is the baseline's training cost over 100x higher than the proposed method's finetuning cost? The paper's own toy experiment (20k iterations vs. 20k iterations) suggests the costs should be identical. Please provide a clear breakdown of this calculation.
2.  Could the authors please clarify the contradiction between the motivation and method? Is the Meanflow model trained on the *couplings from* the 1-rectified flow ($p_{xz}^1$), or is it trained on the *trajectories of* a 2-rectified flow ($v_{\theta^2}$)? If it is the former (as the method section implies), please correct the motivation (Fig 1c, Sec 4.1) which discusses the latter.
3.  To provide a fair comparison, what is the 1-step Euler FID of a standard 2-rectified flow model ($v_{\theta^2}$) that is finetuned from the 1-rectified flow ($v_{\theta^1}$) for the *same* 4 Eflops of compute? This would isolate the benefit of the Meanflow objective versus simply finetuning with the standard Rectified Flow loss.

---

### Official Review · Reviewer_v6jT · 2025-10-27

**Soundness:** 3
**Presentation:** 3
**Contribution:** 3
**Rating:** 4
**Confidence:** 5

**Summary:**

This paper proposes Re-MeanFlow, which trains MeanFlow on flow trajectories straightening via ReFlow. The authors claim that this method combines the benefits of both ReFlow and MeanFlow. Specifically, MeanFlow benefits ReFlow by accounting for curvature in ODE trajectories even after ReFlow, and ReFlow benefits MeanFlow by providing easier (straighter) trajectories and thus stabilizing MeanFlow training. The authors demonstrate the benefits of Re-MeanFlow on toy datasets, ImageNet-64, and ImageNet-256 (latent).

**Strengths:**

- **[S1] The paper is original, as it is the first to combine ReFlow with MeanFlow distillation.** To the best of my knowledge, the most conceptually relevant papers are [1] and [2] which combine ReFlow with direct distillation, i.e., directly regressing neural net output to flow ODE pairs. In contrast to [1] and [2], this paper leverages the fact that flow ODE after ReFlow is still a ODE, so one can apply trajectory distillation techniques such as MeanFlow (one could also apply e.g., Consistency Distillation).

[1] SlimFlow: Training Smaller One-Step Diffusion Models with Rectified Flow, ECCV, 2024

[2] InstaFlow: One Step is Enough for High-Quality Diffusion-Based Text-to-Image Generation, ICLR, 2024

**Weaknesses:**

- **[W1] Incremental techniques.** On the other hand, one could also argue that the techniques proposed in this paper are incremental. Specifically, it is already apparent from the formulation of MeanFlow that one can distill any flow ODE, including ReFlow ODEs, and the idea of improving distillation with ReFlow was explored in [1] with similar motivations. For instance, in page 8 of [1], the authors write "given that the flow is already nearly straight (and hence well approximated by the one-step update), and the distillation can be done efficiently". Furthermore, techniques such as incorporation of CFG was already proposed in the original MeanFlow paper, Gaussian perturbation of $z$ (Eq. (8)) to stabilize ReFlow was explored in [2], and combining real and sync pairs for ReFlow was introduced in [3] and [4].

[1] Flow Straight and Fast: Learning to Generate and Transfer Data with Rectified Flow, 2022.

[2] Balanced Conic Rectified Flow, NeurIPS, 2025.

[3] Improving the Training of Rectified Flows, NeurIPS, 2024.

[4] Simple ReFlow: Improved Techniques for Fast Flow Models, ICLR, 2025.

**Questions:**

- **[Q1] Why does MeanFlow in Figure 4 fail to converge?** I believe the MeanFlow paper also provides strong results on ImageNet-256.

- **[Q2] How is the multi-step generative performance of Re-MeanFlow?** One of the largest benefits of flow maps (in contrast to consistency models whose sample quality degrades with NFE > 2) is their ability to refine sample quality by increasing the number of function evaluation steps [1]. I am curious whether this benefit carries over to Re-MeanFlow.

- **[Q3] Have the authors tried combining ReFlow with AYF or Consistency Models?** Results which show ReFlow also enhances AYF or CM training will strengthen the generality of the paper.

- **[Q4] Have the authors tried applying Re-MeanFlow to more difficult settings such as text-to-image generation?** Given the incremental nature of the proposed techniques, I believe experiments on harder tasks are necessary to improve the significance of the paper.

[1] Align Your Flow: Scaling Continuous-Time Flow Map Distillation, 2025.

---

### Official Review · Reviewer_mDz2 · 2025-10-28

**Soundness:** 2
**Presentation:** 3
**Contribution:** 2
**Rating:** 4
**Confidence:** 4

**Summary:**

The paper proposes  two-stage framework that combines Rectified Flow and MeanFlow for efficient one-step generative modeling.

**- Methods**

The core idea is to first perform a 1-rectified flow to obtain straighter flow trajectories and then fine-tune the pretrained velocity field under the MeanFlow objective (Eq. 5). By training MeanFlow on these rectified trajectories, the authors aim to reduce curvature, stabilize training, and accelerate convergence without requiring multiple rounds of reflow.

The method additionally introduces a noise injection mechanism to handle prior misalignment between inverted latent variables and the Gaussian prior. Each latent vector \( z \) is perturbed as

$z' = \sqrt{1 - \rho}\, z + \sqrt{\rho}\, \epsilon,\ \epsilon \sim \mathcal{N}(0, I),$

where $ \rho \in [0, 1] $ controls the perturbation strength, mitigating collapse and preserving Gaussianity.

**- Results**

Experiments on ImageNet 64² and 256² show that Re-MeanFlow achieves competitive or improved one-step FID scores while consuming substantially fewer FLOPs than 2-Rectified Flow ++ and other distillation methods. A simple 2D Gaussian mixture toy example further illustrates that rectification reduces trajectory curvature, enabling MeanFlow to produce smoother and more stable velocity fields.

**Strengths:**

### **[S1] Practical and Compute-Efficient Design**

Re-MeanFlow offers a low-complexity yet computationally aware improvement, achieving notable efficiency gains without introducing additional architectural or mathematical overhead. By combining a single rectification step with MeanFlow fine-tuning, The method suggests that meaningful reductions in training cost and FLOPs can be achieved while maintaining competitive one-step generation quality.

### **[S2] Adaptability**

The proposed framework is modular and architecture-agnostic, requiring no network modification or retraining from scratch. It can be seamlessly applied as a plug-and-play fine-tuning stage on top of existing pretrained rectified or diffusion models, enabling rapid adaptation to few-step or one-step generation with minimal engineering effort.

**Weaknesses:**

### **[W1] Lack of Theoretical Justification**

- The paper asserts that performing MeanFlow training on once-rectified trajectories leads to smoother average velocity fields and faster convergence, but provides no theoretical or quantitative backing.

- There are no curvature-variance analyses, Lipschitz or Sobolev smoothness bounds, or variance-reduction of velocity fields.

- The supposed “synergy” between rectification and MeanFlow remains qualitative and speculative, rather than formally established, and since the paper’s main contribution hinges on this claimed synergy, additional experimental evidence or theoretical justification is essential to substantiate its validity.

### **[W2] Baseline Fairness**

While the authors report FLOPs including pair-generation, the comparison against original MeanFlow is not fully fair, as Re-MeanFlow benefits from pretrained rectified backbones and an additional generation phase not required by MeanFlow. Moreover, the final FID is slightly worse (3.48 vs 3.43), questioning whether the claimed efficiency gain truly compensates for the added pipeline complexity.

### **[W3] Limited Interpretation of Established Flow Assumptions**

The paper adopts several established principles from prior flow-based studies—such as the benefits of straight ODE trajectories and heuristic noise perturbation for prior alignment—without offering sufficient theoretical interpretation or differentiation.

The claim that straighter ODE trajectories inherently yield better one-step solutions parallels the explanation in [1], which analytically links trajectory straightness with closed-form ODE solvability. However, the present work does not further develop or critically assess this assumption within its own framework.

Similarly, challenges such as DAE weight vanishing and model collapse [2], distribution shift and real-data coupling [3, 4], and perturbation-based supervision [3] have already been explored in earlier works. The proposed method shows methodological similarities—such as reflowed trajectory reuse, Gaussian noise mixing, and hybrid coupling strategies—but lacks a clear theoretical reinterpretation or mechanism that distinguishes it from these prior approaches.

As a result, the paper lacks a clear conceptual bridge between its adopted design choices and the underlying flow principles established in prior work.

### **[W4] Heuristic Noise Injection Without Principle**

- The paper does not link $ \rho $ to any measurable criterion.
- Table 3 presents only four heuristic settings ($ \rho = 0, 0.1, 0.7, \text{Uniform} $) without explaining why $ 0.1 $ performs best or how $ \rho $ could be dynamically adapted.
- The proposed technique closely resembles perturbation-based regularization previously introduced in [3,5] and related works, yet no citation, theoretical grounding, or methodological differentiation is provided.

[1] Rectified Diffusion: Straightness Is Not Your Need in Rectified Flow, 2024.\
[2] Analyzing and Mitigating Model Collapse in Rectified Flow Models, 2024.\
[3] Balanced Conic Rectified Flow, 2025.\
[4] Simple ReFlow: Improved Techniques for Fast Flow Models, 2024.\
[5] SlimFlow: Training Smaller One-Step Diffusion Models with Rectified Flow

**Questions:**

Please refer to the **Weakness** section for detailed comments. It would be helpful if the authors could address those points with further clarification or additional evidence in their rebuttal.

**Details Of Ethics Concerns:**

We have carefully reviewed the ethical implications of this work and did not identify any direct concerns.
The study focuses on improving the computational efficiency of existing flow-based generative models and does not involve human subjects, sensitive data, or deployment of models that could generate harmful or private content.

---

### Official Review · Reviewer_viwV · 2025-10-29

**Soundness:** 3
**Presentation:** 3
**Contribution:** 2
**Rating:** 4
**Confidence:** 5

**Summary:**

This paper proposes to train Meanflow on 2-rectified flow-induced coupling. This reduces gradient variance and thus improves training efficiency.

**Strengths:**

Stabilizing/improving/efficientizing one-step generative models is an important topic.

Motivation sounds.

They provide insightful discussion on the edge cases of the argument made by Lee et al., 2024.

**Weaknesses:**

The method is a combination of 2-rectified flow++ and Meanflow, but it is not clear whether it has a clear benefit over each of them.
- From Table 1, the benefit of Re-Meanflow over Meanflow is not clear, as their FIDs are similar. Is it more efficient? It's hard to see, since EFlops for other methods are not reported in the right sub-table.
- There is no ablation showing the benefit of 2-rf coupling over independent coupling when training Meanflow.
- Lee et al., 2024 used EDM1, not 2. Also, it did not use cfg. Will Re-Meanflow still outperform it if these conditions are matched?



It is not clear from Fig.2 that the 2-rectified flow fails because of the provided reasoning (intersection happening near t=1) or for other reasons. The visualization can be improved.

Fig.2 caption error. (c) is showing Meanflow, not 2-rectified flow. Also, it is not clear what each color means.

**Questions:**

What happens if we use fewer samples than 2.6 * 10^6 (from line 613)? Can we reduce the sampling flops further?

Rather than adding noise to z (Eq 8), can we add noise to x and start ODE solving from a nonzero t0? Adding noise to z may incur more intersections between interpolation paths.

---

### Note · Authors · 2025-11-13

**Comment:**

We sincerely appreciate the reviewers’ insightful and constructive feedback provided on our submission. We are grateful for the recognition of our work’s **clear motivation and illustrations** (e.g., by reviewers viwV, mDz2, and fy6m), **the strength of our empirical results** (fy6m, mDz2), and the **simplicity and plug-and-play nature** of our method (mDz2, v6jT).

At the same time, we fully acknowledge that the suggested additional experiments and analyses are important to further substantiate our claims. Properly addressing these points would require substantial modifications to the paper and significant additional experimentation. Given this, we have decided to withdraw the submission and devote more time to thoroughly improving the work.

We thank the reviewers and the area chair again for their time, effort, and thoughtful evaluation of our paper.

**Withdrawal Confirmation:**

I have read and agree with the venue's withdrawal policy on behalf of myself and my co-authors.